# On Photoeffect in the Few-Electron Atomic Systems

**Alexei M. Frolov** 

Department of Applied Mathematics, University of Western Ontario, London, ON N6H 5B7, Canada; alex1975frol@gmail.com

**Abstract:** Closed analytical formulas are derived for the differential and total cross sections of the non-relativistic photoelectric effect in the three main classes of few-electron atomic systems: (a) neutral atoms and positively charged atomic ions which contain more than one bound electron, (b) negatively charged atomic ions, and (c) one-electron atoms and ions. Our procedure developed in this study is a combination of QED methods and results of the density functional theory obtained for atoms and ions. In all these systems the photoelectric effect is considered as photodetachment of the outer-most electron and our analysis is based on the results of density functional theory obtained for the electron density (radial) distribution in these atomic systems. Analytical formulas (similar to ours) for the differential and total cross sections of photoelectric effect for atomic systems from classes (a) and (b) contribute to our understanding of these systems and have not appeared in the literature, to the best of our knowledge.

**Keywords:** cross sections; photo-effect; photoionization; radial function; photodetachment

## 1. Introduction

In general, the photoelectric effect of atomic system is defined as the absorption of radiation (or photon) by an atom in one of its bound states accompanied by the injection of one of the atomic electrons into a state of unbound spectra. In other words, the bound 'initial' atomic electron makes a transition of into the final unbound state. This unbound (or "free") electron moves away from the parental atom/ion in the Coulomb field of the final, positively charged atomic ion. In the case of negatively charged ions such an unbound photo-electron moves in the field of a neutral atom. Theory of the non-relativistic photoelectric effect (everywhere below, photoeffect, for short), or theory of photoionization of light atoms and ions is a well developed chapter of Quantum Electrodynamics, or QED, for short (see, e.g., [1,2]). In particular, the closed formula for the photoionization cross section of one-electron atoms and ions was produced by Stobbe [3], but it was restricted to the case of ground states in these atomic systems. Since the 1930s, Stobbe's formula was extensively used to explain and describe many actual processes in various physical systems and devices where atomic photoionization plays a crucial role. Twenty years later analogous formula has been derived for the photoionization cross sections from the excited states in one-electron atoms and ions (see, discussion in [4] and references therein). It is a goal of the current paper to show that, within reasonable approximations, quite general formulas can also be derived for the photo-elecric effects in few-electron atomic systems.

In this communication based on a combination of QED approach and results of the density functional theory we report progress toward the solution of these long-standing "unsolvable" problems of atomic photoionization and photodetachment and describe the corresponding analytical solutions. Our main goal in this study is to obtain the closed, analytical formulas for the both differential $\frac{d\sigma}{do}$ and total $\sigma$ cross sections of photoionization and/or photodetachment of few- and many-electron atoms and ions. All these cross sections must be derived as the explicit functions of the atomic ionization potential $I(Q, N_e)$ and $\omega$ is the cyclic frequency of the incident light. The atomic ionization potential $I(Q, N_e)$ is the function of the total number of bounded electrons $N_e$ in the initial atom/ion and

electrical charge $Q$ of the atomic nucleus. To achieve this goal we investigate photoeffect for the outer-most atomic electron, or in other words, photodetachment of the outer-most electron(s) in atomic systems which include few- and many-electron atoms and ions. Below, we shall not consider photoionization of electrons from internal electron shells, double photoionization, photodetachment with instantaneous excitations of other electron transitions and other similar processes. Instead, we will focus on the usual photoionization of that electron which is least weakly bound to the central nucleus. Investigation of such processes is reduced to the analysis of photodetachment of the outer-most electron(s) in atomic system. It is clear that the photodetachment cross sections of the outer-most electron(s) in few-electron atomic systems coincide with the corresponding cross sections of photoionization and/or photodetachment of the whole atom/ion. Expressed concisely, by considering photodetachment of the outer-most electrons in different atomicsystems we can quite accurately describe the regular (or thermal) photoionization and photodetachment in arbitrary few- and many-electron atomic systems.

## 2. Differential Cross Section and Wave Functions

According to the rules of Quantum Electrodynamics (see, e.g., [1,2]) the differential cross section of the non-relativistic photoeffect for an arbitrary atomic system is written in the following form [2]

$$d\sigma = \frac{e^2 m \mid \mathbf{p} \mid}{2\pi\omega} \mid \mathbf{e} \cdot \mathbf{v}_{fi} \mid^2 do = \frac{p}{2\pi\omega a_0} \mid \mathbf{e} \cdot \mathbf{v}_{fi} \mid^2 do \tag{1}$$

where m $\approx$ 9.1093837015 $\times$ 10$^{-28}$g is the rest mass of electron, while $-e$ is its electric charge. Additionally, in this equation $a_0 \approx$ 5.29177210903 $\times$ 10$^{-9}$ cm is the Bohr radius and $\mathbf{e}$ is the vector which describes the actual polarization of initial photon. Furthermore, in this formula $p = \mid \mathbf{p} \mid$ is the momentum of the final (or free) photo-electron, $\omega$ is the cyclic frequency of the incident light quanta, while $\mathbf{v}_{fi}$ is the matrix element of the 'transition' velocity $\mathbf{v}_{fi}$ between initial $|i\rangle$ and final $\langle f|$ states. For this matrix element we can write $\mathbf{v}_{fi} = -\frac{i}{m}\langle\psi_f|\nabla|\psi_i\rangle$, where the notation $\psi$ designates the wave functions, while the indexes $f$ and $i$ in this equation and in all formulas below stand for the final and initial states, respectively. The formula, Equation (1), is written in the relativistic units, where $\hbar = 1, c = 1, e^2 = \alpha$ and $\alpha \approx$ 7.2973525693 $\times$ 10$^{-3}$ ($\approx \frac{1}{137}$) is the dimensionless fine-structure constant. These relativistic units are convenient to perform analytical calculations in Quantum Electrodynamics (below QED, for short). However, in order to determine the non-relativistic cross-sections it is better to apply either the usual units, e.g., *CGS* units, or atomic units in which $\hbar = 1, m_e = 1$ and $e = 1$. In atomic units the formula, Equation (1), takes the form

$$d\sigma = \frac{\alpha a_0^2 p}{2\pi\omega} \mid \mathbf{e} \cdot \langle\psi_f|\nabla|\psi_i\rangle \mid^2 do \quad, \tag{2}$$

where $a_0 = \frac{\hbar^2}{me^2}$ is the Bohr radius and $\alpha = \frac{e^2}{\hbar c}$ is the dimensionless fine-structure constant. In atomic units we have $a_0 = 1$ and $\alpha = \frac{1}{c} (\approx \frac{1}{137})$. Let us assume that initial electron was bound to some atomic system, i.e., to a neutral atom, negatively and/or positively charged ion. The energy of this bound state (or discrete level) is $\varepsilon = -I$, where $I$ is the atomic ionization potential. It is clear that the condition $\omega \geq I$ which must be obeyed to make photoeffect possible. In fact, for atomic photoeffect we always have the following relation $\omega = I + \frac{1}{2}p^2$ between $\omega, I$ and $p$, where $p$ is the momentum of the final photo-electron. This equation is also written in the form $p = \sqrt{2(\omega - I)}$.

### 2.1. Wave Functions of the Final and Initial States

Now, we need to develop some logically closed procedure to calculate the matrix element (or transition amplitude) $\langle\psi_f|\nabla|\psi_i\rangle$ which is included in the formulas, Equations (1) and (2). Everywhere below in this study, we shall assume that the origi-

nal (or incident) atomic system was in its lowest-energy ground (bound) state which is usually $S(L = 0)-$state. For one-electron atomic system this state is always the doublet $1^2 s(\ell = 0)-$state. Then, in the lowest order dipole approximation [1] the outgoing (or final) photo-electron will move in the $p(\ell = 1)-$wave. By using this fact we can write the wave function of the final electron

$$\psi_f(\mathbf{r}) = \frac{2\ell + 1}{2p} \; P_\ell(\mathbf{nn}_1) \; \psi_{\ell;p}(r) = (\mathbf{nn}_1) \; \frac{3}{2p} \psi_{1;p}(r) = (\mathbf{nn}_1) R_{1;p}(r) \,, \tag{3}$$

where $\ell = 1, P_\ell(x)$ is the Legendre polynomial (see, e.g., [5]) and $R_{\ell=1;p}(r) = R_{1;p}(r)$ is the corresponding radial function which depends upon the explicit form of the interaction potential between outgoing photo-electron and remaining atomic system. Furthermore, in this formula the unit vectors $\mathbf{n}$ and $\mathbf{n}_1$ are: $\mathbf{n} = \frac{\mathbf{p}}{p}$ and $\mathbf{n}_1 = \frac{\mathbf{r}}{r}$. The unit vector $\mathbf{n}$ determines the direction of outgoing (or final) photo-electron, or $\mathbf{p}-$direction, for short.

If the interaction potential between outgoing photo-electron and remaining atomic system is described by a Coulomb potential, then the radial function $\psi_{\ell;p}(r)$ in Equation (3) is the normalized Coulomb function of the first kind (see, e.g., [6]) which is

$$\psi_{\ell;p}(r) = \frac{2^\ell Z}{(2\ell + 1)!} \sqrt{\frac{8\pi}{\nu[1 - \exp(-2\pi\nu)]}} \; \prod_{s=1}^{\ell} \sqrt{s^2 + \nu^2} \; (pr)^\ell \; \exp(-\iota pr) \; \times$$
$$_1F_1(\ell + 1 + \iota\nu, 2\ell + 2; 2\iota pr) \,, \tag{4}$$

where $_1F_1(a, b; z)$ is the confluent hypergeometric function defined exactly as in [5,6], $\nu = \frac{Z}{p} = \frac{Q - N_e + 1}{p}$ and $Z = Q - N_e + 1$ is the electric charge of the final atomic fragment. From this equation one finds the wave function of an electron which moves in a central Coulomb field in the $p-$wave (i.e., for $\ell = 1$)

$$\psi_{1;p}(r) \;\; = \;\; \frac{2Zp}{3!} \sqrt{\frac{8\pi(1 + \nu^2)}{\nu[1 - \exp(-2\pi\nu)]}} \; r \exp(-\iota pr) \; _1F_1(2 + \iota\nu, 4; 2\iota pr) \,. \tag{5}$$

By multiplying this formula by the additional factor $\frac{3}{2p}$ from Equation (3), we can write for this wave function with $\ell = 1$ (in atomic units)

$$
\begin{aligned}
R_{1;p}(r) \;\; &= \;\; Z \sqrt{\frac{2\pi(1 + \nu^2)}{\nu\left(1 - \exp(-2\pi\nu)\right)}} \; r \exp(-\iota pr) \; _1F_1(2 + \iota\nu; 4; 2\iota pr) \\
&= \;\; p \sqrt{\frac{2\pi\nu(1 + \nu^2)}{1 - \exp(-2\pi\nu)}} \; r \exp(-\iota pr) \; _1F_1(2 + \iota\nu; 4; 2\iota pr) \,, 
\end{aligned} \tag{6}
$$

where the electric charge $Z$ of the remaining atomic system is an increasing function of the nuclear charge $Q$, but it also depends upon the total number of bound electrons $N_e$. All phases and normalization factors in this formula coincide exactly with their values presented in [7].

For the non-Coulomb (or short-range) interaction potentials between outgoing photo-electron and remaining atomic core, the normalized radial wave function of the continuous spectra is written as a product of the spherical Bessel function $j_\ell(pr)$ and a factor which equals $2p$, i.e., $\psi_{\ell;p}(r) = 2pj_\ell(pr)$ (see, e.g., [7], $\S$ 33). For $\ell = 1$ one finds $\psi_{1;p}(r) = 2pj_1(pr)$. This means that the function $\psi_{\ell=1;p}(r)$ is regular at $r = 0$. From here for the radial function $R_{1;p}(r) = \frac{3}{2p}\psi_{\ell=1;p}(r)$ we obtain

$$R_{1;p}(r) = \frac{3}{2p}\Big[2pj_{\ell=1}(pr)\Big] = 3j_{\ell=1}(pr) = 3\sqrt{\frac{\pi}{2pr}} \; J_{\frac{3}{2}}(pr) \,, \tag{7}$$

where $j_1(x) = \frac{sinx}{x^2} - \frac{cosx}{x}$ and $J_{\frac{3}{2}}(x)$ is the Bessel function which is regular at the origin (at $r = 0$) and defined exactly as in [5]. These two radial wave functions of an unbound photo-electron, Equations (6) and (7), are used in our calculations below.

Now, let us discuss the wave functions of the initial atomic system which has one nucleus with the electric charge $Q$ and $N_e$ bound electrons. By analyzing the current experimental data it is easy to understand that the non-relativistic photodetachment of the outer-most electrons in a few- and many-electron atomic systems is produced by photons with large and very large wavelengths $\lambda$. In reality, the wavelengths $\lambda$ of incident light quanta substantially exceed the actual sizes of atoms and ions which are included in this process. For instance, the wavelengths $\lambda$ of photons that produce photodetachment of the negatively charged hydrogen ions $H^-$ in Solar photosphere exceed 7000 Å $\approx$ 13,232 *a.u.*, while the spatial radius $R$ of this ion equals $\approx 2.710\ a_0 = 2.710\ a.u.$, i.e., $R \ll \lambda$. In fact, for the $H^-$ ion its spatial radius $R$ is smaller than the wavelengths $\lambda$ of incident light quanta in thousands times. In other words, photodetachment of the outer-most electron in the $H^-$ ion is produced at large and very large distances from the central atomic nucleus and from the second atomic electron. Such asymptotic spatial areas in the $H^-$ ion are very important to determine photodetachment cross sections, since only in these spatial areas one finds a relatively large overlap between the electron and photon wave functions.

Similar situations can be found in other atomic systems considered in this study, e.g., for all neutral atoms and positively charged ions. In each of these Coulomb systems photodetachment of the outer-most electron(s) mainly occurs in the asymptotic areas of their wave functions. These asymptotic areas are located far and very far form the central atomic nucleus and other internal atomic electrons. Therefore, in our analysis of non-relativistic photodetachment of the outer-most electron(s) we can restrict ourselves to large spatial areas and consider only the long-range asymptotic of these wave functions. Moreover, it seems very tempting to neglect the small area of electron-electron correlations around the central nucleus ($R \leq a_0$) and consider the long-range asymptotics of atomic wave functions as the 'new' wave functions for our problem. Briefly, in this procedure we replace the actual wave functions for each of these atomic systems by their long-range radial asymptotics. It is clear that the 'new' bound state wave function is one-electron function and it has a different normalization constant. Obviously, this is an approximation, but as follows from our results the overall accuracy of our approximation is very good and sufficient to describe photodetachment of the outer most-electrons in all atomic systems discussed in this study.

In this study, we consider the photoeffect in the three different classes of atomic systems: (a) atom/ion which initially contains $N_e$ bound electrons, while its nuclear charge $Q$ ($Q \geq N_e$) is arbitrary, (b) one-electron atom/ion, where $N_e = 1$ and $Q$ is arbitrary, and (c) negatively charged ion where $Q = N_e - 1$ and the both $Q$ and $N_e$ are arbitrary. As is well known (see, e.g., Equations (3.12) and (3.13) in Ref. [8] and references therein) in arbitrary atomic ($Q, N_e$)-system the radial wave function of the ground $S(L = 0)$-states has the following long-distance (radial) asymptotics

$$
\begin{aligned}
R_i(r) &= C(b;I)r^{b-1}\exp(-\sqrt{2I}r) = \frac{(2\sqrt{2I})^{b+\frac{1}{2}}}{2\sqrt{\pi\Gamma(2b+1)}}\ r^{b-1}\ \exp(-\sqrt{2I}r) \\
&= \frac{(2\sqrt{2I})^{b+\frac{1}{2}}}{2\sqrt{\pi\Gamma(2b+1)}}\ r^{\frac{Q-N_e+1}{\sqrt{2I}}-1}\ \exp(-\sqrt{2I}r) \qquad (8)
\end{aligned}
$$

where $b = \frac{Q-N_e+1}{\sqrt{2I}} = \frac{Z}{\sqrt{2I}}$, $I (\geq 0)$ is the atomic ionization potential and $Z = Q - N_e + 1$ is the electric charge of the remaining atom/ion. In Equation (8) and everywhere below the notation $\Gamma(x)$ denotes the Euler's gamma-function $\Gamma(1 + x) = x\Gamma(x)$, which is often called the Euler's integral of the second kind [5]. This important result of the Density Functional Theory (or DFT, for short) plays a crucial role in this study. Here we have to emphasize the following fundamental fact: the formula, Equation (8), is the exact long-range asymptotic

of the truly correlated, $N_e$-electron wave function of an actual atom/ion. The derivation of this formula is not based on any approximation. In other words, by choosing the wave function $R_i(r)$ in the form of Equation (8) we do not neglect any of the electron-electron correlations in atomic wave function.

However, since photodetachment of the outer-most electrons mainly occurs at large and very large distances from the atomic nucleus, then it will be a very good approximation to describe this phenomenon, if we continue the radial $R_i(r)$ function, Equation (8), on the whole real $r$-axis, including the radial origin, i.e., the point $r = 0$. This allows us to determine the factor $C(b; I)$ in Equation (8) which is the normalization constant of the radial $R_i(r)$ function, which now continues on the whole real $r$-axis, including the radial origin, i.e., the point $r = 0$. Namely, after this step our analysis becomes approximate. Nevertheless, we can determine the normalization constant $C(b; I)$ for this radial $R_i(r)$ function, Equation (8) where $0 \leq r < +\infty$. It equals

$$C(b; I) = \frac{(2\sqrt{2I})^{b+\frac{1}{2}}}{2\sqrt{\pi\Gamma(2b+1)}} \tag{9}$$

where the atomic ionization potential $I$ and parameter $b$ are the two real, non-negative numbers. In some equations below the $\sqrt{2I}$ value is also designated as $B$. In the general case, the atomic ionization potential $I$ is an unknown function of $Q$ and $N_e$.

For the negatively charged (atomic) ions we always have $b = \frac{Q-N_e+1}{\sqrt{2I}} = 0$. The long-distance asymptotic of the radial wave function of an arbitrary negatively charged ion is always written in the form: $R(r) \sim \frac{C}{r}\exp(-\sqrt{2I}r)$, where $C = \sqrt[4]{\frac{I}{2\pi^2}}$ is the normalization constant and $I$ is an unknown function of $Q$ and $N_e$. In contrast with this, for one-electron atoms and ions we have $N_e = 1$ and $2I = Q^2$ and ionization potential $I$ is the uniform function of the nuclear charge $Q$ only. For one-electron atomic systems we also have $b = 1$ and the exact wave function is written in the form $R(r; Q) = A\exp(-\sqrt{2I}r)$, where $2I = Q^2$, and $A = \frac{Q\sqrt{Q}}{\sqrt{\pi}}$ is the normalization constant.

### 2.2. Gradient Operator and Its Matrix Elements

Let us derive some useful formulas for the matrix element $\langle \psi_f | \boldsymbol{\nabla} | \psi_i \rangle$ which is included in Equation (2) and plays a central role in this study. It is clear that we need to determine the vector-derivative (or gradient) of the initial wave function, which is a scalar function. In general, for the interparticle (or relative) vector $\mathbf{r}_{ij} = \mathbf{r}_j - \mathbf{r}_i$ the corresponding gradient operator in spherical coordinates takes the form (see, e.g., [9])

$$\nabla_{ij} = \frac{d}{d\mathbf{r}_{ij}} = \frac{\mathbf{r}_{ij}}{r_{ij}}\frac{\partial}{\partial r_{ij}} + \frac{1}{r_{ij}}\nabla_{ij}(\Omega) = \mathbf{e}_{r;ij}\frac{\partial}{\partial r_{ij}} + \mathbf{e}_{\theta;ij}\frac{1}{r_{ij}}\frac{\partial}{\partial \theta_{ij}} + \mathbf{e}_{\phi;ij}\frac{1}{r_{ij}\sin\theta_{ij}}\frac{\partial}{\partial \phi_{ij}}, \tag{10}$$

where $\nabla_{ij}(\Omega)$ is the angular part of the gradient vector which depends upon angular variables ($\theta$ and $\phi$) only, while $\mathbf{e}_{r;ij} = \frac{\mathbf{r}_{ij}}{r_{ij}} = \mathbf{n}_{ij}$, $\mathbf{e}_{\theta;ij}$ and $\mathbf{e}_{\phi;ij}$ are the three unit vectors in spherical coordinates which are defined by the $\mathbf{r}_j$ and $\mathbf{r}_i$ vectors, where $\mathbf{r}_j \neq \mathbf{r}_i$.

If the radial part of the initial wave function depends upon the scalar radial variable only, then all derivatives in respect to the both angular variables $\theta$ and $\phi$ equal zero identically and we can write

$$\boldsymbol{\nabla}_{ij}R(r) = \frac{\mathbf{r}_{ij}}{r_{ij}}\frac{\partial R(r_{ij})}{\partial r_{ij}} = \frac{\mathbf{r}_{ij}}{r_{ij}}\frac{dR(r_{ij})}{dr_{ij}} = \mathbf{n}_{ij}\frac{dR(r_{ij})}{dr_{ij}}, \tag{11}$$

where $\mathbf{n}_{ij}$ is the unit vector in the direction of interparticle $\mathbf{r}_{ij}$ variable. For one-center atomic systems we can determine $r_{1j} = r_j$, and for one-electron systems $r_{12} = r_1 = r$. In this case,

the formula, Equation (11), is written in the form: $\nabla R(r) = \mathbf{n}_1 \frac{dR(r)}{dr}$, where $\mathbf{n}_1 = \frac{\mathbf{r}_1}{r_1}$. In this notation the radial matrix element is

$$
\mathbf{e}\langle \psi_f | \nabla | \psi_i \rangle = \int_0^{+\infty} \left\{ \oint (\mathbf{n} \cdot \mathbf{n}_1)(\mathbf{e} \cdot \mathbf{n}_1) do_1 \right\} \left( R_{1;p}(r) \frac{dR_i(r)}{dr} \right) r^2 dr
$$

$$
= \frac{4\pi}{3} (\mathbf{e} \cdot \mathbf{n}) \int_0^{+\infty} \left( R_{1;p}(r) \frac{dR_i(r)}{dr} \right) r^2 dr = \frac{4\pi}{3} (\mathbf{e} \cdot \mathbf{n}) I_{rd} \ , \tag{12}
$$

where $\mathbf{n}$ is the unit vector which determines the direction of propagation of the final electron, while $R_{1;p}(r)$ and $R_i(r)$ are the radial functions of the final and initial states, respectively. The notation $I_{rd}$ in this formula, Equation (12), stands for the following auxiliary radial integral

$$
I_{rd} = \int_0^{+\infty} \left( R_{1;p}(r) \frac{dR_i(r)}{dr} \right) r^2 dr = - \int_0^{+\infty} \left( R_i(r) \frac{dR_{1;p}(r)}{dr} \right) r^2 dr \ , \tag{13}
$$

where we used the so-called 'transfer of the derivative' (or partial integration) which often helps to simplify analytical calculations of this radial integral.

By substituting the expression, Equation (12), into the formula, Equation (2), one finds the following 'final' formula for the differential cross section of the non-relativistic photodetachment of an arbitrary atomic system

$$
d\sigma = \frac{16\pi^2 \alpha a_0^2 p}{18\pi \omega} (\mathbf{e} \cdot \mathbf{n})^2 \mid I_{rd} \mid^2 do = \frac{8\pi \alpha a_0^2 p}{9\omega} (\mathbf{e} \cdot \mathbf{n})^2 \mid I_{rd} \mid^2 do \ . \tag{14}
$$

As follows from this formula the angular distribution of photo-electrons is determined by the 'angular' factor $(\mathbf{e} \cdot \mathbf{n})^2$. This cross section of photodetachment corresponds to the truly (or 100 %) polarized light. However, in many actual applications the incident beam of photons is unpolarized and we deal with the natural (or white) light. If the incident beam of photons was unpolarized, then we need to apply the formula $\overline{(\mathbf{e} \cdot \mathbf{n})^2} = \frac{1}{2}(\mathbf{n}_l \times \mathbf{n})^2$, where $\mathbf{n}_l$ is the unit vector which describes the direction of incident light propagation and $\mathbf{n}$ is the unit vector which determines the direction of propagation of the final photo-electron. In this study, the notation $(\mathbf{a} \times \mathbf{b})$ denotes the vector product of the two vectors $\mathbf{a}$ and $\mathbf{b}$. Finally, the differential cross section of photodetachment is written in the form

$$
d\sigma = \left( \frac{4\pi \alpha a_0^2 p}{9\omega} \right) (\mathbf{n}_l \times \mathbf{n})^2 \mid I_{rd} \mid^2 do = \frac{4\pi}{9} \alpha a_0^2 \left( \frac{p}{\omega} \right) \sin^2 \Theta \mid I_{rd} \mid^2 do \ , \tag{15}
$$

where $\Theta$ is the angle between two unit vectors $\mathbf{n}_l$ and $\mathbf{n}$. The presence of vector product $(\mathbf{n}_l \times \mathbf{n})^2$ in Equation (15) is typical for the dipole approximation. As follows from the formula, Equation (15), analytical and numerical calculations of the differential cross section of photodetachment are now reduced to analytical computations of the auxiliary radial integral $I_{rd}$, Equation (13). The total cross section of photodetachment is

$$
\sigma = \left( \frac{32\pi^2 \alpha a_0^2 p}{27\omega} \right) \mid I_{rd} \mid^2 = \frac{32\pi^2 \alpha a_0^2}{27} \left( \frac{p}{\omega} \right) \mid I_{rd} \mid^2 \ . \tag{16}
$$

By using different expressions for the initial and final wave functions we can determine the differential and total cross sections of photodetachment of the outer most electrons in various few- and many-electron atoms and ions. The corresponding formulas are presented below.

## 3. Few-Electron Neutral Atoms and Positively Charged Ions

First, let us consider photodetachment of the outer-most electron(s) in few-electron neutral atoms, where $Q = N_e$ and $N_e \geq 2$, and in positively charged atomic ions, where $Q > N_e$ and $N_e \geq 2$. In both these cases the final sub-systems, i.e., outgoing photo-electron and remaining positively charged ion, interact with each other by an attractive Coulomb

potential. For atoms and positively charged ions this process obviously coincides with atomic photoionization. The wave function of outgoing photo-electron must be taken in the form of Equation (6), while the wave function of the initial atomic state is chosen in the form of Equation (8). The radial derivative of this initial wave function is

$$\frac{d}{dr}\left[r^{b-1}\exp(-Br)\right] = (b-1)r^{b-2}\exp(-Br) - Br^{b-1}\exp(-Br) \ , \tag{17}$$

where $b = \frac{Q-N_e+1}{\sqrt{2I}} = \frac{Z}{\sqrt{2I}} = \frac{Z}{B}, Z = Q - N_e + 1$ and $B = \sqrt{2I}$. Therefore, the formula for our auxiliary radial integral $I_{rd}$, Equation (13), includes the two terms $I_{rd} = I_{rd}^{(1)} + I_{rd}^{(2)}$, where

$$I_{rd}^{(1)} = p\sqrt{\frac{2\pi v(1+v^2)}{1-\exp(-2\pi v)}}C(b;B)(b-1)\int_0^{+\infty} r^{(b+2)-1}\exp(-Br-\imath pr)_1F_1(2+\imath v;4;$$

$$2\imath pr)dr = p\sqrt{\frac{2\pi v(1+v^2)}{1-\exp(-2\pi v)}}C(b;B)\frac{(b-1)\Gamma(b+2)}{(B+\imath p)^{b+2}}\ {}_2F_1\left(2+\imath v;b+2;4;\frac{2\imath p}{B+\imath p}\right) \tag{18}$$

$$= p\sqrt{\frac{2\pi v(1+v^2)}{1-\exp(-2\pi v)}}C(b;B)\frac{(b-1)\Gamma(b+2)}{(B+\imath p)^{b+2}}\left(\frac{B+\imath p}{B-\imath p}\right)^{\imath v+b}\ {}_2F_1\left(2-\imath v;2-b;4;\frac{B-\imath p}{B+\imath p}\right) \ ,$$

where $_2F_1(a,b;c;z)$ is the (2,1)-hypergeometric function defined exactly as in [5], $v = \frac{Z}{p}$ and $Z = Q - N_e + 1$, while $C(b;B)$ is the normalization constant of the bound state radial function (see, Equation (9)). The explicit formula for the second radial integral $I_{rd}^{(2)}$ is

$$I_{rd}^{(2)} = -p\sqrt{\frac{2\pi v(1+v^2)}{1-\exp(-2\pi v)}}C(b;B)\ B\int_0^{+\infty} r^{(b+3)-1}\exp(-Br-\imath pr)_1F_1(2+\imath v;4;$$

$$\imath 2pr)dr = -p\sqrt{\frac{2\pi v(1+v^2)}{1-\exp(-2\pi v)}}C(b;B)\frac{B\ \Gamma(b+3)}{(B+\imath p)^{b+3}}\ {}_2F_1\left(2+\imath v;b+3;4;\frac{2\imath p}{B+\imath p}\right) \tag{19}$$

$$= -p\sqrt{\frac{2\pi v(1+v^2)}{1-\exp(-2\pi v)}}C(b;B)\frac{B\ \Gamma(b+3)}{(B+\imath p)^{b+3}}\left(\frac{B+\imath p}{B-\imath p}\right)^{\imath v+b+1}\ {}_2F_1\left(2-\imath v,1-b;4;\frac{B-\imath p}{B+\imath p}\right) \ .$$

where $C(b;B)$ is the normalization constant, Equation (9), and $B = \sqrt{2I}$, where $I$ is the ionization potential of the initial atomic systems. To simplify the two last formulas we note that

$$\left(\frac{B+\imath p}{B-\imath p}\right)^{\imath v} = \left[\left(\frac{\frac{v}{b}+\imath}{\frac{v}{b}-\imath}\right)^{\imath \frac{v}{b}}\right]^b = \exp\left[-2v\ \text{arccot}\left(\frac{v}{b}\right)\right] \tag{20}$$

where $v = \frac{Z}{p} = \frac{Q-N_e+1}{p}$ and $b = \frac{Q-N_e+1}{\sqrt{2I}}$ (here $Z = Q - N_e + 1$, see above) and $\frac{B}{p} = \frac{v}{b}$, or $v = \frac{bB}{p}$. The function arccot $x$ is the inverse cotangent function which is also equals arccot $x = \arccos(\frac{x}{\sqrt{1+x^2}})$ (this formula is used in numerical calculations).

After a few additional, relatively simple transformations we derive to the following expression for the total radial integral $I_{rd} = I_{rd}^{(1)} + I_{rd}^{(2)}$:

$$I_{rd} = \frac{p^2}{b}\sqrt{\frac{v(1+v^2)}{1-\exp(-2\pi v)}}\frac{2^b B^b \sqrt{B}\Gamma(b+2)(B-\imath p)^{1-b}}{\sqrt{\Gamma(2b+1)(B^2+p^2)^2}}\exp\left[-2v\ \text{arccot}\left(\frac{v}{b}\right)\right]\left[(b-1)\times\right.$$

$$(v-\imath b)\ {}_2F_1\left(2-\imath v;2-b;4;\frac{v-\imath b}{v+\imath b}\right) - v\ (b+2)\ {}_2F_1\left(2-\imath v,1-b;4;\frac{v-\imath b}{v+\imath b}\right)\bigg] \ . \tag{21}$$

From this expression one easily finds the following formula for the $\mid I_{rd}\mid^2$ value

$$| I_{rd} |^2 = \frac{4^b\, v^{2b+2}(1+v^2)\Gamma^2(b+2)}{p(v^2+b^2)^{b+3}(1-\exp(-2\pi v))\Gamma(2b+1)}\, \exp\left[-4v\,\text{arccot}\left(\frac{v}{b}\right)\right]\left|(b-1)\times\right.$$

$$\left.(v-\imath b)\; {}_2F_1\left(2-\imath v; 2-b; 4; \frac{v-\imath b}{v+\imath b}\right) - v\,(b+2)\, {}_2F_1\left(2-\imath v, 1-b; 4; \frac{v-\imath b}{v+\imath b}\right)\right|^2. \tag{22}$$

By multiplying this expression by the $\left(\frac{8\pi\alpha a_0^2 p}{9\omega}\right)$ $(\mathbf{n}\cdot\mathbf{e})^2$ factor one finds the final formula for the differential cross section of photoionization of few- and many-electron neutral atoms and/or positively charged ions each of which contains $N_e$ bound electrons ($N_e \geq 2$) and one central atomic nucleus with the electrical charge $Q$

$$d\sigma = \left(\frac{8\pi\alpha a_0^2}{9\omega}\right)\frac{4^b\, v^{2b+2}(1+v^2)\Gamma^2(b+2)}{(v^2+b^2)^{b+3}(1-\exp(-2\pi v))\Gamma(2b+1)}\, \exp\left[-4v\,\text{arccot}\left(\frac{v}{b}\right)\right]\left|(b-1)\times\right.$$

$$\left.(v-\imath b)\; {}_2F_1\left(2-\imath v; 2-b; 4; \frac{v-\imath b}{v+\imath b}\right) - v\,(b+2)\, {}_2F_1\left(2-\imath v, 1-b; 4; \frac{v-\imath b}{v+\imath b}\right)\right|^2 \tag{23}$$

$$(\mathbf{n}\cdot\mathbf{e})^2 do\,,$$

where the incident beam of light is completely polarized. For natural light we obtain the following formula

$$d\sigma = \left(\frac{4\pi\alpha a_0^2}{9\omega}\right)\frac{4^b\, v^{2b+2}(1+v^2)\Gamma^2(b+2)}{(v^2+b^2)^{b+3}(1-\exp(-2\pi v))\Gamma(2b+1)}\, \exp\left[-4v\,\text{arccot}\left(\frac{v}{b}\right)\right]\left|(b-1)\times\right.$$

$$\left.(v-\imath b)\; {}_2F_1\left(2-\imath v; 2-b; 4; \frac{v-\imath b}{v+\imath b}\right) - v\,(b+2)\, {}_2F_1\left(2-\imath v, 1-b; 4; \frac{v-\imath b}{v+\imath b}\right)\right|^2 \times \tag{24}$$

$$(\mathbf{n}_l \times \mathbf{n})^2 do\,.$$

Finally, the corresponding formula for the total cross section of photoionization of the $(Q, N_e)-$atomic system, where $N_e \geq 2$, takes the form

$$\sigma = \left(\frac{32\pi^2\alpha a_0^2}{27\omega}\right)\frac{4^b\, v^{2b+2}(1+v^2)\Gamma^2(b+2)}{(v^2+b^2)^{b+3}(1-\exp(-2\pi v))\Gamma(2b+1)}\, \exp\left[-4v\,\text{arccot}\left(\frac{v}{b}\right)\right] \times \tag{25}$$

$$\left|(b-1)(v-\imath b)\; {}_2F_1\left(2-\imath v; 2-b; 4; \frac{v-\imath b}{v+\imath b}\right) - v\,(b+2)\, {}_2F_1\left(2-\imath v, 1-b; 4; \frac{v-\imath b}{v+\imath b}\right)\right|^2.$$

Note that the parameter $b$ in these formulas is a real number, which is usually bounded between 0 and 2, i.e., $0 < b < 2$. This means that the hypergeometric functions in Equations (18) and (19) can be determined only numerically (two exceptional cases when $b = 1$ and $b = 0$ are considered in the next two Sections). Recently, a number of fast, reliable and numerically stable algorithms have been developed and tested for accurate calculations of the hypergeometric functions. Our final formula can be simplified even further, if one applies the following relation (see, e.g., Eq.15.2.3 in [6]) between two hypergeometric functions which are included in our Equation (22):

$${}_2F_1\left(2-\imath v, 2-b; 4; \frac{v-\imath b}{v+\imath b}\right) = {}_2F_1\left(2-\imath v, 1-b; 4; \frac{v-\imath b}{v+\imath b}\right)$$

$$+ \frac{1}{1-b}\left(\frac{v-\imath b}{v+\imath b}\right)\frac{d}{dz}\left[{}_2F_1(2-\imath v, 1-b; 4; z)\right]\,, \tag{26}$$

where $z = \frac{v-\imath b}{v+\imath b}$. This formula allows one to operate with one hypergeometric function only.

All formulas derived and presented in this Section can directly be used to determine the both differential and total cross sections of photodetachment of the outer-most electrons in few- and many-electron neutral atoms and positively charged ions which contains $N_e$ bound electron, where $N_e \geq 2$. Note also that the total cross section of photoionization of the

$(Q, N_e)$−atomic system, Equation (25) can be re-written in a slightly different form $\sigma(\omega) = \sigma(\omega; I, b)$ which shows the explicit dependence of the photoionization cross section upon the cyclic frequency of incident light $\omega$. Such explicit formulas are very popular among people which conduct experiments and those theorists who calculate the convolution of various energy spectra, e.g., thermal spectra, and photodetachment cross sections. To achieve this goal one needs to replace variables in Equation (25) by using the following (equivalent) expression for $v$ written in terms of $I, \omega$ and $b$: $v = b\sqrt{\frac{I}{\omega - I}}$. Finally, one finds the following formula for the total cross section $\sigma = \sigma(\omega, I, b)$ of photodetachment of the outer-most electrons in few-electron neutral atoms and positively charged ions

$$
\begin{aligned}
\sigma =& \left(\frac{32\pi^2 \alpha a_0^2}{27}\right) \frac{4^b I^{b+2} \left[1 + (b^2 - 1)\frac{I}{\omega}\right] \Gamma^2(b+3)}{b^4 \omega^{b+3} \left[1 - \exp\left(-2\pi b\sqrt{\frac{I}{\omega - I}}\right)\right] \Gamma(2b+1)} \exp\left[-4b\sqrt{\frac{I}{\omega - I}} \operatorname{arccot}\left(\sqrt{\frac{I}{\omega - I}}\right)\right] \\
&\times \left| \frac{b-1}{b+2} \left(1 - \imath b\sqrt{\frac{\omega - I}{I}}\right) {}_2F_1\left(2 - \imath b\sqrt{\frac{I}{\omega - I}}, 2 - b; 4; \frac{\sqrt{I} - \imath\sqrt{\omega - I}}{\sqrt{I} + \imath\sqrt{\omega - I}}\right) \right. \\
&- \left. {}_2F_1\left(2 - \imath b\sqrt{\frac{I}{\omega - I}}, 1 - b; 4; \frac{\sqrt{I} - \imath\sqrt{\omega - I}}{\sqrt{I} + \imath\sqrt{\omega - I}}\right) \right|^2 .
\end{aligned}
\tag{27}
$$

This formula is one of the main results of this study. For one-electron atomic systems, when $b = 1$, the last formula, Equation (27), takes a familiar form:

$$
\begin{aligned}
\sigma =& \left(\frac{32\pi^2 \alpha a_0^2}{27}\right) \frac{4 I^3 \Gamma^2(4)}{\omega^4 \left[1 - \exp\left(-2\pi\sqrt{\frac{I}{\omega - I}}\right)\right] \Gamma(3)} \exp\left[-4\sqrt{\frac{I}{\omega - I}} \operatorname{arccot}\left(\sqrt{\frac{I}{\omega - I}}\right)\right] \\
=& \frac{256\pi^2 \alpha a_0^2}{3} \frac{I^3}{\omega^4 \left[1 - \exp\left(-2\pi\sqrt{\frac{I}{\omega - I}}\right)\right]} \exp\left[-4\sqrt{\frac{I}{\omega - I}} \operatorname{arccot}\left(\sqrt{\frac{I}{\omega - I}}\right)\right],
\end{aligned}
\tag{28}
$$

which exactly coincides with another our formula, Equation (39), derived below (Stobbe's formula). Indeed, for one-electron atomic systems we have $b = 1$ and $I = \frac{Q^2}{2}$, and therefore, $\frac{I^3}{\omega^4} = \left(\frac{I}{\omega}\right)^4 \frac{2}{Q^2}$. This is the first known derivation of the formula, Equation (39), (or Stobbe's formula) for one-electron atom and ions from our much more general and universal expression, Equation (28), which is also applicable for arbitrary few- and many-electron atoms and positively charged ions. Earlier studies on related problems (see Ref. [4] and earlier references therein) did not result in general closed analytical expression for the photoionization cross sections in few- and many-electron atoms/ions; Equation (28) reports progress in this direction.

The both formulas, Equations (24) and (27), can be re-written in slightly different forms, if we introduce the universal photoionization function $F_+(b, x) = F_+(b, \frac{I}{\omega})$, which is defined for few- and many-electron atoms and positively charged atomic ions by the equation:

$$
\begin{aligned}
F_+(b, x) =& \frac{4^b \Gamma(b+3)}{b^4 \Gamma(2b+1)} \frac{x^{b+3}[1 + (b^2 - 1)x]}{1 - \exp(-2\pi b y)} \times \\
& \left| \frac{b-1}{b+2} \left(1 - \imath\frac{b}{y}\right) {}_2F_1\left(2 - \imath\frac{b}{y}, 2 - b; 4; \frac{y - \imath}{y + \imath}\right) - {}_2F_1\left(2 - \imath\frac{b}{y}, 1 - b; 4; \frac{y - \imath}{y + \imath}\right) \right|^2 ,
\end{aligned}
\tag{29}
$$

where $x = \frac{I}{\omega}$, $y = \sqrt{\frac{x}{1-x}}$ and $b = \frac{Q - N_e + 1}{\sqrt{2I}}$. By using this universal photoionization function $F_+(b, x)$ we can write the following, compact expressions for the differential and total cross sections

$$\frac{d\sigma}{do} = \left(\frac{4\pi\alpha a_0^2}{9}\right) F_+\left(b, \frac{I}{\omega}\right) (\mathbf{n}_l \times \mathbf{n})^2 \quad \text{and} \quad \sigma(\omega) = \left(\frac{32\pi^2\alpha a_0^2}{27}\right) F_+\left(b, \frac{I}{\omega}\right) \ . \tag{30}$$

## 4. One-Electron Atoms and Ions

Photoionization of one-electron atomic systems is significantly simpler than photodetachment of the outer-most electrons in a few- and many-electron atomic systems considered above. Indeed, in this case we have the atomic ionization potential $I$ which depends upon the nuclear charge $Q$ only, i.e., $I = I(Q)$. Furthermore, for the ground (bound) state in one-electron atoms and ions we always have $2I = Q^2$, and therefore, in the both formulas, Equations (18) and (19) the parameter $b = 1$ and the first term in Equation (17) equals zero identically. Furthermore, for $b = 1$ the hypergeometric functions $_2F_1(2 - \imath\nu, 0; 4; z)$, which is included in the second term, equals unity. This means that for one-electron atoms/ions (or for $b = 1$) we can express the both differential and total cross sections of photoionization in terms of elementary functions only. The normalization constant of the incident wave function equals $C(b; B) = C(1; Q) = \frac{Q\sqrt{Q}}{\sqrt{\pi}}$ and the two parameters $\nu$ and $\frac{\nu}{b}$ are now identical. This means that for any one-electron atom and/or ion the long-range asymptotics of its actual wave function always coincides with the wave function itself, and for ground states it is also coincides with the formula, Equation (8). In other words, by applying our method to the ground states in one-electron atomic systems we obtain the exact solution which correctly describes the non-relativistic photoionization (or photoeffect).

First, we note that for one-electron atoms and ions the explicit formula for the auxiliary radial integral $I_{rd}$ in Equation (12) (or $I_{rd}^{(2)}$ in Equation (17)) takes the form

$$\begin{aligned} I_{rd} &= -pQ^2 \frac{Q\sqrt{Q}}{\sqrt{\pi}} \sqrt{\frac{2\pi(1+\nu^2)}{\nu(1 - \exp(-2\pi\nu))}} \int_0^{+\infty} r^{(4-1)} \exp(-Qr - \imath pr) {}_1F_1(2 + \imath\nu; 4; 2\imath pr) dr \\ &= -pQ^3\sqrt{Q} \sqrt{\frac{2(1+\nu^2)}{\nu(1 - \exp(-2\pi\nu))}} \frac{\Gamma(4)}{(Q + \imath p)^4} \ {}_2F_1\left(2 + \imath\nu; 4; 4; \frac{2\imath p}{Q + \imath p}\right) , \end{aligned} \tag{31}$$

where $\nu = \frac{Q}{p}$ and $\Gamma(4) = 3 \cdot 2 \cdot 1 = 6$. The hypergeometric function in the last equation can be transformed to the form

$$_2F_1\left(2 + \imath\nu; 4; 4; \frac{2\imath p}{Q + \imath p}\right) = \left(1 - \frac{2\imath p}{Q + \imath p}\right)^{4 - 4 - 2 - \imath\nu} {}_2F_1\left(2 - \imath\nu; 0; 4; \frac{2\imath p}{Q + \imath p}\right) = \left(\frac{Q + \imath p}{Q - \imath p}\right)^{2 + \imath\nu} , \tag{32}$$

where we applied the formula $_2F_1(\alpha, \beta; \gamma; z) = (1 - z)^{\gamma - \alpha - \beta} {}_2F_1(\gamma - \alpha, \gamma - \beta; \gamma; z)$ (see, Equation (9.131) in [5]). Another way to obtain the same formula, Equation (32), is to apply the following expression for the integral in Equation (31) with the confluent hypergeometric function(s) (see, e.g., [5,7])

$$\int_0^{+\infty} \exp(-\lambda z) z^{\gamma - 1} {}_1F_1(\alpha; \gamma; bz) dz = \frac{\Gamma(\gamma)}{\lambda^\gamma} \left(\frac{\lambda}{\lambda - b}\right)^\alpha \ . \tag{33}$$

Finally, we obtain

$$
\begin{aligned}
I_{rd} &= -6Q^3\sqrt{Q}\sqrt{\frac{2(1+\nu^2)}{\nu(1-\exp(-2\pi\nu))}}\,\frac{1}{(Q^2+p^2)^2}\left(\frac{\nu+\imath}{\nu-\imath}\right)^{\imath\nu} \\
&= -\frac{6Q^3\sqrt{Q}}{p^4}\sqrt{\frac{2}{\nu(1-\exp(-2\pi\nu))\,(1+\nu^2)^3}}\,\frac{\exp(-2\nu\,\mathrm{arccot}\,\nu)}{(1+\nu^2)^2}\ .
\end{aligned}
\tag{34}
$$

By using the notation $\nu = \frac{Q}{p}$ and multiplying the radial integral $I_{rd}$ by the factor $\frac{4\pi}{3}(\mathbf{n}\cdot\mathbf{e})$ we derive the following formula

$$
\frac{4\pi}{3}(\mathbf{n}\cdot\mathbf{e})I_{rd} = -\frac{8\pi}{\sqrt{p}}\sqrt{\frac{2\nu^6}{[1-\exp(-2\pi\nu)](1+\nu^2)^3}}\,\exp(-2\nu\,\mathrm{arccot}\,\nu)(\mathbf{n}\cdot\mathbf{e})\ .
\tag{35}
$$

Now, the explicit formula for the $\frac{p\alpha a_0^2}{2\pi\omega}\mid I_{rd}\mid^2$ factor in Equation (2) takes the form

$$
\frac{p\alpha a_0^2}{2\pi\omega}\mid I_{rd}\mid^2 = \frac{64\pi\alpha a_0^2}{\omega}\left(\frac{\nu^2}{1+\nu^2}\right)^3\frac{\exp(-4\nu\,\mathrm{arccot}\,\nu)}{1-\exp(-2\pi\nu)}\ .
\tag{36}
$$

The final formula for the differential cross sections of photoionization of one-electron atom/ion by a completely polarized light is written in the form

$$
\begin{aligned}
d\sigma &= \frac{64\pi\alpha a_0^2}{\omega}\left(\frac{\nu^2}{1+\nu^2}\right)^3\frac{\exp(-4\nu\,\mathrm{arccot}\,\nu)}{1-\exp(-2\pi\nu)}(\mathbf{n}\cdot\mathbf{e})^2 do \\
&= \frac{128\pi\alpha a_0^2}{Q^2}\left(\frac{I}{\omega}\right)^4\frac{\exp(-4\nu\,\mathrm{arccot}\,\nu)}{1-\exp(-2\pi\nu)}(\mathbf{n}\cdot\mathbf{e})^2 do\ ,
\end{aligned}
\tag{37}
$$

where we also used the following relation $1 = \frac{Q^2}{Q^2} = 2\left(\frac{Q^2}{2}\right)\frac{1}{Q^2} = \frac{2I}{Q^2}$. For the natural (or unpolarized) light the differential cross section of photoionization is written in the form

$$
d\sigma = \frac{64\pi\alpha a_0^2}{Q^2}\left(\frac{I}{\omega}\right)^4\frac{\exp(-4\nu\,\mathrm{arccot}\,\nu)}{1-\exp(-2\pi\nu)}(\mathbf{n}\times\mathbf{n}_l)^2 do\ .
\tag{38}
$$

The total cross section of photoionization for one-electron atom/ion with the nuclear charge $Q$ is

$$
\begin{aligned}
\sigma &= 512\pi^2\alpha\left(\frac{a_0^2}{Q^2}\right)\left(\frac{I}{\omega}\right)^4\frac{\exp(-4\nu\,\mathrm{arccot}\,\nu)}{1-\exp(-2\pi\nu)} \\
&= 512\pi^2\alpha\left(\frac{a_0^2}{Q^2}\right)\left(\frac{I}{\omega}\right)^4\frac{\exp\left(-4\sqrt{\frac{I}{\omega-I}}\,\mathrm{arccot}\,\sqrt{\frac{I}{\omega-I}}\right)}{1-\exp\left(-2\pi\sqrt{\frac{I}{\omega-I}}\right)}\ .
\end{aligned}
\tag{39}
$$

This formula can also be re-written in the form

$$
\sigma = \frac{256\pi^2\alpha a_0^2}{I}\left[\frac{x^4\,\exp(-4\,y\,\mathrm{arccot}\,y)}{1-\exp(-2\pi\,y)}\right] = \frac{256\pi^2\alpha a_0^2}{I}\,F_Q(x)\ ,
\tag{40}
$$

where $x = \frac{I}{\omega}, y = \sqrt{\frac{x}{1-x}}$ and $F_Q(x)$ is the universal photoionization function defined for one-electron atom(s) and positively charged ions in which the nuclear charge of the central nucleus equals $Q$.

Our formulas, Equations (38)–(40), exactly coincide with the formula obtained by Stobbe in [3] and with the analogous formulas derived in § 56 from [2]. Note also that the last formulas, Equations (37)–(40), have directly been derived from our formula, Equation (27),

obtained in the previous Section for $b = 1$, but here we wanted to derive and check them by using an independent approach. As mentioned above for one-electron atoms and ions its ionization potential $I$ is the explicit (and simple) function of the nuclear charge $Q$ only. Generalization of our formulas to photodetachment of the outer-most electron from the excited atomic states is also simple and transparent, but it cannot be done directly with the use of DFT theory, since Equation (8) does not work for the excited states. Derivation of the explicit formulas for photoionization cross sections of the excited one-electron atoms and ions also requires additional explanations, extra notations and extensive analytical work.

## 5. Negatively Charged Ions

Investigation of the non-relativistic photoeffect in the negatively charged ions is reduced to the analysis of photodetachment of the outer-most electron in similar atomic systems. Note that photodetachment of the negatively charged atomic ions is of great interest in numerous applications. In general, there is a fundamental difference in photodetachment of the negatively charged ions and photoionization of the positively charged atomic ions and neutral atoms. In particular, for all negatively charged ions the electrical charge of the final atom $Z = Q - N_e + 1$ equals zero identically. Therefore, in this case we cannot introduce the parameter $\nu = \frac{Z}{p}$, which was very helpful in the two previous Sections. This means that all our formulas, derived for the photodetachment cross sections (see below), contain only the momentum of photo-electron $p$ and ionization potential $I$, or parameter $B = \sqrt{2I}$. These two variables $p$ and $I$ (or $B$) are crucial for theoretical analysis of the non-relativistic photoeffect in the negatively charged ions, or their photodetachment. Such a photodetachment of the negatively charged, two-electron H$^-$ ion is of great interest for our understanding of all details in actual visible and infrared spectra of many stars, including our Sun. Photodetachment of the four-electron negatively charged Li$^-$ ion plays some role in developing of very compact and reliable photo-elements and recharged batteries. Therefore, it is important to produce some universal formula for the photodetachment cross sections of the negatively charged ions.

For the negatively charged atomic ions the derivative of the radial wave function of the initial state, Equation (8), is written in the form

$$\frac{d}{dr}\left[\frac{C}{r}\exp(-Br)\right] = -Cr^{-2}\exp(-Br) - CBr^{-1}\exp(-Br) , \tag{41}$$

where $C$ is the normalization constant which equals $\sqrt[4]{\frac{I}{2\pi^2}}$ as follows from Equation (9) for $b = 0$. Therefore, the formula for our auxiliary radial integral $I_{rd}$, Equation (13), will also include two different terms, i.e., $I_{rd} = 3\left(J_{rd}^{(1)} + J_{rd}^{(2)}\right)$, where

$$J_{rd}^{(1)} = C\sqrt{\frac{\pi}{2p}}\int_0^{+\infty} dr J_{\frac{3}{2}}(pr)r^{\frac{1}{2}-1}\exp(-Br) = C\sqrt{\frac{\pi}{2p}}\left(\frac{p}{2}\right)^{\frac{3}{2}}\frac{\Gamma(2)}{\Gamma\left(\frac{5}{2}\right)(B^2+p^2)} \times \tag{42}$$

$$_2F_1\left(1,1;\frac{5}{2};\frac{p^2}{B^2+p^2}\right) = \frac{Cp}{3(B^2+p^2)}\; _2F_1\left(\frac{3}{2},\frac{3}{2};\frac{5}{2};\frac{p^2}{B^2+p^2}\right) .$$

The hypergeometric function in the last equation can be reduced to some combination of elementary functions. To show this explicitly let us apply the following formula

$$\frac{d}{dz}\left[_2F_1(\alpha,\beta;\gamma;z)\right] = \frac{\alpha\beta}{\gamma}\; _2F_1(\alpha+1,\beta+1;\gamma+1;z) , \tag{43}$$

where in our case $\alpha = \frac{1}{2}, \beta = \frac{1}{2}$ and $\gamma = \frac{3}{2}$. For these values of $\alpha, \beta$ and $\gamma$ the last formula takes the form

$$\frac{d}{dz}\left[_2F_1\left(\frac{1}{2},\frac{1}{2};\frac{3}{2};z\right)\right] = \frac{1}{6}\; _2F_1\left(\frac{3}{2},\frac{3}{2};\frac{5}{2};z\right) , \tag{44}$$

where the argument $z$ varies between zero and unity, i.e., $0 \leq z < 1$. In our case this is true, since $z = \frac{p^2}{B^2 + p^2}$. Now, we can write

$$_2F_1\left(\frac{3}{2}, \frac{3}{2}; \frac{5}{2}; z\right) = 6\frac{d}{dz}\left[_2F_1\left(\frac{1}{2}, \frac{1}{2}; \frac{3}{2}; z\right)\right] = 6\frac{d}{dz}\left(\frac{\arcsin\sqrt{z}}{\sqrt{z}}\right) = 3\frac{\sqrt{z} - \sqrt{1-z}\,\arcsin\sqrt{z}}{z\sqrt{z(1-z)}} \ , \tag{45}$$

where we used the formula Equation (15.1.6) from [6] for the $_2F_1\left(\frac{1}{2}; \frac{1}{2}; \frac{3}{2}; z\right)$ hypergeometric function, i.e., $_2F_1\left(\frac{1}{2}; \frac{1}{2}; \frac{3}{2}; z\right) = \frac{\arcsin\sqrt{z}}{\sqrt{z}}$. The analytical formula, Equation (45), derived for this $_2F_1\left(\frac{3}{2}, \frac{3}{2}; \frac{5}{2}; z\right)$ function is our original result which cannot be found directly neither in [5], nor in [6]. In our case $z = \frac{p^2}{B^2 + p^2}$, $\sqrt{1-z} = \frac{B}{\sqrt{B^2 + p^2}}$ and $\sqrt{z} = \frac{p}{\sqrt{B^2 + p^2}}$ and the final formula for the $J_{rd}^{(1)}$ integral takes the form

$$J_{rd}^{(1)} = \frac{C}{\sqrt{B^2 + p^2}}\ \frac{\sqrt{z} - \sqrt{1-z}\,\arcsin\sqrt{z}}{z\sqrt{1-z}} \ . \tag{46}$$

The expression in the right-hand side of this equation is not singular when $z \to 0$ (or $p \to 0$), since

$$\lim_{z \to 0}\frac{\sqrt{z} - \sqrt{1-z}\,\arcsin\sqrt{z}}{z\sqrt{1-z}} = \frac{1}{6}\lim_{z \to 0}\sqrt{z} = 0 \ .$$

Analogous formula for the second radial integral $J_{rd}^{(2)}$ is

$$J_{rd}^{(2)} = C\,B\sqrt{\frac{\pi}{2p}}\int_0^{+\infty}dr J_{\frac{3}{2}}(pr)r^{\frac{3}{2}-1}\exp(-Br) = \frac{C}{\sqrt{2p}}\left(\frac{p}{2}\right)^{\frac{3}{2}}\frac{B\,\Gamma(3)}{\Gamma\left(\frac{5}{2}\right)(B^2 + p^2)^{\frac{3}{2}}} \times$$

$$_2F_1\left(\frac{3}{2}; \frac{1}{2}; \frac{5}{2}; \frac{p^2}{B^2 + p^2}\right) = \frac{2\,C\,B\,p}{3(B^2 + p^2)^{\frac{3}{2}}}\ _2F_1\left(\frac{3}{2}; \frac{1}{2}; \frac{5}{2}; \frac{p^2}{B^2 + p^2}\right) . \tag{47}$$

It is possible to obtain the explicit expression for the $_2F_1\left(\frac{3}{2}; \frac{1}{2}; \frac{5}{2}; z\right)$ hypergeometric function in terms of some elementary functions. For this purpose we need to use the known analytical formula for the $_2F_1\left(\frac{1}{2}, \frac{1}{2}; \frac{3}{2}; z\right)$ function (which equals $\frac{\arcsin\sqrt{z}}{\sqrt{z}}$, see, Equation (45)) and apply the formula Equation (15.2.7) from [6] for $n = 1$ which takes the form

$$\frac{d}{dz}\left[(1-z)^a {}_2F_1(a, b; c; z)\right] = -\frac{a(c-b)}{c}(1-z)^{a-1}{}_2F_1(a+1, b; c+1; z) , \tag{48}$$

where $a = \frac{1}{2}, b = \frac{1}{2}$ and $c = \frac{3}{2}$. Now, for the $_2F_1\left(\frac{1}{2}, \frac{1}{2}; \frac{3}{2}; z\right)$ function one finds

$$\frac{d}{dz}\left[(1-z)^{\frac{1}{2}}{}_2F_1\left(\frac{1}{2}, \frac{1}{2}; \frac{3}{2}; z\right)\right] = -\frac{1}{3}(1-z)^{-\frac{1}{2}}\ _2F_1\left(\frac{3}{2}, \frac{1}{2}; \frac{5}{2}; z\right) . \tag{49}$$

From this equation we derive

$$_2F_1\left(\frac{3}{2}, \frac{1}{2}; \frac{5}{2}; z\right) = -3\sqrt{1-z}\frac{d}{dz}\left[\sqrt{1-z}\left(\frac{\arcsin\sqrt{z}}{\sqrt{z}}\right)\right] = \frac{3}{2}\left(\frac{\arcsin\sqrt{z}}{\sqrt{z}}\right.$$

$$-\ \frac{\sqrt{1-z}}{z} + \frac{1-z}{z\sqrt{z}}\arcsin\sqrt{z}\right) = \frac{3}{2\sqrt{z}}\left(\frac{\arcsin\sqrt{z}}{z} - \frac{\sqrt{z(1-z)}}{z}\right) , \tag{50}$$

where $z = \frac{p^2}{B^2 + p^2} < 1$ and $z \geq 0$. This analytical formula for the $_2F_1\left(\frac{3}{2}, \frac{1}{2}; \frac{5}{2}; z\right)$ function is another original result which cannot be found neither in [5], nor in [6]. Note also that

analytical formula for the integral in Equation (47) can also be derived as a partial derivative of the $J_{rd}^{(1)}$ in respect to the parameter $B$. Thus, the both auxiliary radial integrals $J_{rd}^{(1)}$ and $J_{rd}^{(2)}$ are expressed in terms of the elementary functions. In particular, the final formula for the $J_{rd}^{(2)}$ integral is

$$J_{rd}^{(2)} = \frac{C}{\sqrt{B^2 + p^2}} \sqrt{1-z} \left[ \frac{\arcsin \sqrt{z}}{z} - \sqrt{\frac{1-z}{z}} \right], \tag{51}$$

where $z = \frac{p^2}{B^2+p^2}$, $\sqrt{z} = \frac{p}{\sqrt{B^2+p^2}}$ and $\sqrt{1-z} = \frac{B}{\sqrt{B^2+p^2}}$. Again, by using the formula, Equation (1.641) from [5] for the $\arcsin x$ one can easily show that

$$\lim_{z \to 0} \left( \frac{\arcsin \sqrt{z}}{z} - \sqrt{\frac{1-z}{z}} \right) = 0 ,$$

which means that our formula for the $J_{rd}^{(2)}$ integral is not singular at $z \to 0$ (or at $p \to 0$). Note that the formula Equation (2) for the differential cross section of photodetachment always contains an additional factor $p$ in its numerator. Therefore, such a cross-section for an arbitrary negatively charged ion always approaches zero when $p \to 0$. The same statement is true for the total cross sections of photodetachment of the negatively charged ions. In contrast with this, analogous cross sections (differential and total) of atomic systems considered in the two previous Sections approach (when $p \to 0$) some final limits which are not equal zero. All these features of photoionization cross sections of the neutral atoms and positively charged ions are well known from numerous experiments and theoretical calculations (see, e.g., [4,10] and references therein).

Analytical computations of the total auxiliary $I_{rd} = 3\left( J_{rd}^{(1)} + J_{rd}^{(2)} \right)$ integral and the both differential and total cross sections are simple and straightforward. The final formula, Equation (15), for the differential cross section of photodetachment of the negatively charged atomic ions takes the form

$$d\sigma = \left( \frac{8\pi\alpha a_0^2 p}{9\omega} \right) (\mathbf{n} \cdot \mathbf{e})^2 \mid I_{rd} \mid^2 do = \frac{8\alpha a_0^2}{\omega} \left[ \frac{pB}{\omega(B^2+p^2)} \right] \mid J_{rd}^{(1)} + J_{rd}^{(2)} \mid^2 (\mathbf{n} \cdot \mathbf{e})^2 do \tag{52}$$

for completely polarized light. Analogous formula for unpolarized light is

$$d\sigma = \frac{4\alpha a_0^2}{\omega} \sqrt{\frac{1-z}{z}} \left[ (\sqrt{1-z} - 1) \frac{\arcsin(\sqrt{z})}{\sqrt{z}} + \frac{1}{\sqrt{1-z}} - 1 + z \right]^2 (\mathbf{n}_l \times \mathbf{n})^2 do, \tag{53}$$

where $z = \frac{p^2}{B^2+p^2}$, $\sqrt{z} = \frac{p}{\sqrt{B^2+p^2}}$, $\sqrt{1-z} = \frac{B}{\sqrt{B^2+p^2}}$ and $B^2 = 2I$. The formula for the total cross section is written in the form

$$
\begin{aligned}
\sigma &= \frac{32\pi\alpha a_0^2}{3\omega} \sqrt{\frac{1-z}{z}} \left[ (\sqrt{1-z} - 1) \frac{\arcsin(\sqrt{z})}{\sqrt{z}} + \frac{1}{\sqrt{1-z}} - 1 + z \right]^2 \\
&= \frac{32\pi\alpha a_0^2}{3I} \sqrt{\frac{x}{1-x}} \left[ (\sqrt{x(1-x)} - 1) \frac{\arcsin \sqrt{1-x}}{\sqrt{1-x}} + 1 - x\sqrt{x} \right]^2 = \frac{32\pi\alpha a_0^2}{3I} F_-(x)
\end{aligned}
\tag{54}
$$

where $z = 1 - \frac{I}{\omega}$, $x = 1 - z = \frac{I}{\omega}$ and $0 \le x < 1$. The function $F_-(x) = F_-\left( \frac{I}{\omega} \right)$ defined in this equation is the universal photodetachment function which is applied to an arbitrary negatively charged ion. Derivation of the formulas for the differential and total cross sections of photodetachment of the negatively charged atomic ions is one of the main results of this study. The $F_-(x)$ function defined in Equation (54) is the same for all negatively charged ions which means that the cross sections of photodetachment for all

negatively charged ions are similar (in this sense) to each other. Briefly, this means that there is no principal difference between photodetachment cross sections of the $H^-$, $Li^-$ and $O^-$ ions. It can be shown that the $F_-(x)$ function has one maximum in the area of our interest, i.e., for $0 \leq x < 1$. The both amplitude and location of this maximum on the $\omega$ axis depend upon the ionization potential $I$ only. The differential cross section of photodetachment of an arbitrary negatively charged ion is also represented in a simple analytical form with the universal $F_-(x) = F_-\left(\frac{I}{\omega}\right)$ function.

The formulas presented in this Section allow one to describe (completely and accurately) photoeffect in the negatively charged atomic ions. Derivation of our formulas for the differential and total cross sections of photodetachment of the negatively charged atomic ions is one of the main results of this study. All these formulas contain only elementary functions, and this was a real scientific surprise. Note also that our formula, Equation (54), for the $\sigma = \sigma(\omega, I)$ dependence allows one to check and mainly confirm some earlier predictions made by Chandrasekhar in their papers about photodetachment cross-section of the negatively charged $H^-$ ion [11,12] (see also discussion in Section 74 of [4,13,14] and references therein). The formulas derived above can also be used to describe photodetachment of the weakly-bound deuterium nucleus [15].

## 6. Discussion and Conclusions

In this study, by using the methods of quantum electrodynamics, we have developed the closed analytical procedure to describe the photoelectric effect in few-electron atoms and positively charged ions, as well as in the negatively charged atomic ions. The electron density distributions in the incident atoms/ions have been taken from DFT [8]. Based on this procedure we have studied the non-relativistic photoeffect in different atomic systems including the neutral atoms and positively charged atomic ions which contain $N_e$ bound electrons ($N_e \geq 2$) in an atom/ion with the nuclear charge $Q$. Photoionization of one-electron atoms/ions and photodetachment of the negatively charged atomic ions, where $N_e = Q + 1$ (or $Q = N_e - 1$), are also investigated. In each of these cases we have derived the closed analytical formulas for the both differential and total cross sections of photoionization and/or photodetachment (see, Equations (23), (25), (53) and (54) above) of the corresponding atoms and ions. Note that each of these formulas contains only a few basic parameters of the original problem, e.g., the cyclic frequency of incident light $\omega$, atomic ionization potential $I$, the total number of initially bound electrons $N_e$ and electrical charge of atomic nucleus $Q$. These and some other formulas are the main results of this study. For neutral atoms and positively charged atomic ions with $N_e \geq 2$ and for negatively charged atomic ions similar formulas have never been produced in earlier papers. Our procedure developed here allows one to determine (both analytically and numerically) the differential and total cross sections of thermal photoionization of arbitrary few- and many-electron atomic systems. Our results obtained for the neutral He atom and $Li^+$ ion agree very well with the results of previous numerical calculations [16,17] of these systems. Maximal deviations of our total cross sections of photoeffect for these two systems and similar cross sections obtained in [16,17] do not exceed 8–10%.

Our formulas derived for few-electron atoms and positively charged ions have been tested in applications to one-electron atoms and ions. Note that our analytical expressions for the differential and total cross sections of one-electron atomic systems have been derived in the both ways, i.e., directly and as a limit (when $N_e \rightarrow 1$) of the formulas obtained for photoionization of neutral atoms and positively charged ions. Remarkably, but all these formulas coincide with each other and with the well known formulas obtained earlier in [2,3]. Furthermore, our analytical formulas derived for the photodetachment cross sections (differential and total) of the negatively charged ions are original and include only elementary functions. None of these formulas has ever been produced in earlier studies. By using our formulas we have determined the differential and total cross sections of the negatively charged $H^-$ ion which are in good agreement with fundamental calculations performed in [10] (see, also [16] and references therein). For the negatively charged ions

our differential and total cross sections for the negatively charged ions deviate from the results of numerical computations [10] does not exceed 18%.

Thus, the qualitative and quantitative agreement of our formulas with the known computational results for differential and total cross sections confirms the consistency of our approach. This is true for all three classes of few-electron atomic systems, i.e., for all few-electron neutral atoms and ions, for the negatively charged ions and for one-electron atoms and ions. Currently, similar cross sections are determined either in numerical computations, or experimentally. However, from the columns of numbers with many digits in each it is very hard (even impossible) to derive the corresponding analytical formulas which produce these results.

Recently, there is an increasing interest to perform highly accurate computations of the both photoionization and photodetachment cross sections for the light few-electron atoms and ions. Probably, very soon similar calculations of the cross sections will be performed by taking into account the lowest order relativistic and QED corrections. In general, the knowledge of differential and total ionization cross sections is crucial for the description of the quantum dynamics (line shape) in high-precision atomic experiments (see, e.g., [18,19]). Typically, some of the sophisticated techniques employed in recent experiments, such as two-photon spectroscopy, can lead to transitions into the continuum, which need to be taken into account in the analysis of the experiments. Results of such experiments can be used to improve the numerical values of some fundamental atomic constants, including the fine-structure constant, Rydberg constant, etc. Based on these results we can also determine some crucial components of the lowest order QED correction(s), e.g., the Lamb shifts, for those atomic systems [18] where alternative methods do not work properly.

**Funding:** This research received no external funding.

**Institutional Review Board Statement:** Not applicable.

**Informed Consent Statement:** Not applicable.

**Data Availability Statement:** Not applicable.

**Conflicts of Interest:** The author declares no conflict of interest.

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
