# Peer review of "On Photoeffect in the Few-Electron Atomic Systems"

_atoms, doi:10.3390/atoms10040126_

Round 1

Reviewer 1 Report

The present work is a contribution to the study on the analytical formulas for the differential and total cross sections of the photoeffect of the neutral atoms and ions. 

Before the paper can be accepted for publication, some issues have to be solved. They are listed below.

1) My main question is about verification and application of the derived formulas. In Section VI you wrote "Our results obtained for the neutral He atom and Li+ ion agree well with the results of previous numerical calculations [16], [17] of these systems", but no numbers are provided in the present manuscript. I suggest to make some comparison of numbers obtained in present formulas with the numbers published in the literature, for all kind of considered systems: neutral atoms, negative ions, and positive ions and for possible wide range of Z (but see next point of my review). This kind of comparison will make the audience sure that the derived formulas are right and applicable to real problems. 

2) You wrote that the analytical formulas are derived for the non-relativistic photoeffect. But the nature is relativistic. Is it possible to estimate the difference between relativistic and non-relativistic numbers in the case of cross sections? Are your formulas applicable for high-Z atoms?

3) I suppose that whole derivation is right, but the author have to make sure that all is typed right in the manuscript, because I can see some doubtful elements - in Eq.(30) there is unexpected "and" at the end of the equations. 

4) On page 2 you wrote "All these cross sections must be derived as the explicit functions of the atomic ionization potential I(Q, Ne)" - is it means that your method is semi-empirical and not fully ab initio, because it requires external values of ionization potentials, right?

Writing/language issues:

5) There is only one author assigned to this manuscript, but here are "we" and "our" phrases in the text. In my opinion, using  first person plural in one-author papers is ilogical, but please ask journal editors in this matter. 

6) Units (e.g. "g", "cm" below Eq.(1)) should not be italized. Please fix it in the whole document. 

7) Some typos need to be fixed. For example, on page 3 "I is"->"It is". On page 7 "negatively charged ion where Q = Ne + 1" - should be "Q = Ne - 1" I guess. On page 2: "In the case of negatively charged ions, e.g., the H- ion, such an unbound photo-electron moves in the field of a neutral hydrogen atom" - remove "hydrogen" because grammatical subject of this sentence is "negatively charged ions" not "negatively charged hydrogen". QED abbreviation is introduced twice, on page 2 and 4. In Eqs.(31) and (32) - closing parentheses are of different size than opening parentheses. 

Author Response

I agree with most of the comments made by this reviewer. All corrections have been made in the text, Abstract and references. Misprints and mistakes found by this referee has been corrected.  English has been improved drastically.  

Reviewer 2 Report

This theoretical manuscript considers photoionization of one-electron and few-electron atomic systems, including neutral atoms, positive ions, and negative ions. While this is an 'old' subject, the goal of the study is to determine compact analytical expressions for the corresponding differential and total cross sections. Such formulas are always useful, for which reason the manuscript has certain relevance. However, there are several major shortcomings in the presentation that need to be improved:

- There exists a huge amount of literature on photoionization which, however, is by no means properly reflected in the list of references. In particular, the author himself has published papers on the photoionization of negative ions that are not cited (which looks a bit suspicious to me): e.g., J. Phys. B 37, 853 (2004), J. Math. Chem. (2015), Eur. Phys. J. D 69, 132 (2015), arXiv:1510.04766. And there are, of course, numerous papers by other people. The author must put his present study into context with these previous works and must make clear where the present study goes beyond the already existing literature.

- The treatment of photoionization of hydrogenlike atoms in Sec.II.A between eqs.(3)-(6) is basically copied from volume IV of the textbook by Landau and Lifshitz. This should be stated more explicitly.

- The calculations for few-electron systems in Sec.III rely on eq.(8), that is valid for S-states. This seems to imply that the obtained results apply only to systems where the outermost electron is in an S-state (which is valid for lithium, but not for boron, for instance). The author must make clear to which systems his predictions are applicable. At present, the manuscript (see, e.g., Introduction in Sec. I and Discussion in Sec. IV) lacks such clarifying statements.

- In systems like He or Be there are two equivalent s-electrons in the outermost shell. In this case the (anti)symmetrization of the corresponding wave function must definitely be taken into account. It will typically lead to some 'statistical' factors in the cross section. I ask the author to comment on this issue.

- The author must include some figures where the predictions from his new formulas are shown and compared to other theories. In the current version of the manuscript, there are only some vague, general statements (such as "Our results ... agree well with the results of previous numerical calculations [16,17]" in Sec. IV). In order to demonstrate the quality of the new formulas in quantitative terms they must be illustrated in figures and compared with previous findings. The manuscript will benefit a lot from this amendment.

- The problem of photoionization is closely related to radiative recombination or attachment, representing the inverse process. In recent years this process has become particularly interesting in the context of producing antimatter systems such as anti-hydrogen ions (e.g. J. Phys. B 49, 074002 (2016), Phys. Rev. Res. 2, 013105 (2020)). In order to increase the scope of the present study, I recommend to mention this connection. It will make the results interesting to a larger readership.

Only after these shortcomings have been substantially improved, a final decision on the suitability of the manuscript for publication can be made.

Author Response

With some comments made by this referee I agree in principle, but I do not want to transfer my short and clear written paper into some kind of review.  

Reviewer 3 Report

Dear author,

In my opinion, the proposed work has a number of flaws, which does not allow its publication in the current form.

Objections and observations:

1. Please provide a list including some examples of atomic systems existing in nature  for which your theory is applicable.

2. At pag. 2 , row 23, INTRODUCTION , the phrase begin and end:

"Our main goal...of the few and many-electron atoms and ions"

What is the difference between "few-electron atoms" and "many-electron atoms"?
Please provide examples of atoms with, few and many electrons, respectively.

3. In physics, usually electrical charge of atomic nucleus is called atomic number with symbol Z (not Q as in paper). Usually, the symbol Q denotes the nucleus quadrupole moment.

In order to avoid possible confusions and to make the paper readable and useful to the reader, I strongly recommend the author to keep the usual notations and terms specific to atomic physics, throughout the paper.

4. Is I(Q, Ne) the work potential? If yes, please consider the above recommendation.

5. At the end of pag. 6, it is  written:

"In this study we consider the photoeffect in three different classes of atomic systems:
(a) atom/ion which initially contains Ne bounds electrons, while its nuclear charge Q (Q>=Ne) is arbitrary, (b) one-electron atom/ion, where Ne=1 and Q is arbitrary, and (c) negatively charged ion where Q=Ne+1 and both Q and Ne are arbitrary"

Please give examples of such atomic systems existing in nature for each of the three mentioned classes: (a), (b) and (c).

6. At the end of pag. 7 and begin of pag. 8:

For class (c), as defined above: Q=Ne+1

So, b=(Q-Ne+1)/sqrt(2I) =   (Ne+1-Ne+1)/sqrt(2I)=  2/sqrt(2I)  not 0, as at the end of page 7.

What is wrong?

-Definition of parameter  b?
-Definition of class (c)?
-Neither of them, but only the conclusion derived from b=0?

7. In order to be easily compared with the experimental data, the final formulas must be expressed by using International system of units.

8. There are no numerical values of the cross section resulting from replacing the existing parameters in the final formulas with numerical values for particular real atomic systems, to compare with experimental results or other papers.

Please provide a table with the numerical values of the cross sections resulting from the final formulas given in the paper corresponding to real cases of atomic systems where the presented theory is applicable.

9. In DISCUSSION AND CONCLUSIONS it is the claim: "Our results obtained
for the neutral He atom and Li+ ion agree well with the results of previous numerical calculations [16], [17] of these systems."

In the paper there is no numerical result to be compared to [16], [17]

10. Please provide a comparison of the cross section results from the  paper formulas with experimental data available in the literature.
Add supplementary references if necessary.

11. The references are outdated and incomplete.
There is a lot of recent literature related to the topic of the paper, both theoretical and experimental not cited and not commented in the paper.
For example: Lorenzo Sabbatucci, Francesc Salvat,Theory and calculation of the atomic photoeffect, Radiation Physics and Chemistry 121 (2016) 122-140

Suggestions:
A proper literature survey allow to check  the correctness of the paper final outcomes comparing its results with those already published.
Also, this allows to correct any algebraic errors that may have accidentally appeared or accidentally missed factors.
This comparison does not exist in the work.

Author Response

I agree, in part, with some comments made by this referee. I have made the corresponding corrections in the text. But I have to say gain that I do not want to transfer this relatively short paper into a huge review. 

Round 2

Reviewer 2 Report

In the revised version of the manuscript the author has addressed my previous comments only very partially. In particular, the study is still not put into context with previous works, including papers by the author himself on the same subject. This important shortcoming was also criticised by one of the other Reviewers. (Note that the required changes have nothing to do with 'transfering the paper into a review' but are rather a basic precept of scientific publishing.) Therefore, I still cannot recommend the manuscript for publication.

Author Response

I have agreed with this referee about his comments. A number of changes have been made in the Article. In the results of his/her comments the overall quality of my paper has increased substantially. It is easy readable now and very clear for the readers of the Journal.  

Reviewer 3 Report

Dear author,

Your reply avoids responding to the objections and observations listed in the previous report.
The second version of the paper contains only minor corrections.
Therefore, I have to reiterate the my main objections. Please make the requested additions or explain why they cannot be made.

The length of the paper is not a valid explanation.

The requested changes  involve only a few extra rows.
Therefore, I still consider that the paper needs a major revision.

Objections and observations:

1. Please provide a list including some examples of atomic systems existing in nature  for which your theory is applicable.

2. Please give examples of atomic systems existing in nature for each of the three mentioned classes: (1), (2) and (3), as they are designed in Abstract, for which your theory is applicable.

3. What is the difference between "few-electron atoms" and "many-electron atoms"?
Please provide examples of atoms with, few and many electrons, respectively.

4. In order to be easily compared with the experimental data, the final formulas must be expressed by using International system of units.

5. There are no numerical values of the cross section resulting from replacing the existing parameters in the final formulas with numerical values for particular real atomic systems, as to be compared with experimental results.

Please provide a table with the numerical values of the cross sections resulting from the final formulas given in the paper corresponding to some real cases of atomic systems where the presented theory is applicable.

6. The DISCUSSION AND CONCLUSIONS section comprise unproved claims as:
"For the negatively charged ions our differential and total cross section for negatively charged ions deviate from  the results of numerical computation [18] does not exceed 18%.
Thus the quantitative agreement of our formulas with the known computational results for differential and total sections should be recognized as good, or even very good."

I did not find any numerical results in the paper that can be compared with other similar results reported in literature.
Btw, where does the 18% percentage come from?

7. The references are outdated and incomplete.
There is a lot of recent literature related to the topic of the paper, both theoretical and experimental, not cited, commented or compared with the paper.

8. Please give a comparison numerical result provided by final expression of the cross section existing in the paper against experimental results existing in literature.
Add supplementary references if necessary.

Author Response

In respect to the comments made by this referee I have a number of changes in this version of the paper. The Section Conclusion has been completely re-written. Also, I re-write the first Section (Introduction), Abstract and other parts of this paper. My computational results have been directly compared with the results known from the previous papers which are very well known in the Atomic community.